# Novel Cellular and Immunotherapy: Toxicities and Perioperative Implications

Alessandro De Camilli and Gregory Fischer *

Memorial Sloan Kettering, 1275 York Avenue, New York, NY 10065, USA
* Correspondence: fischerg@mskcc.org

**Abstract:** Targeted cellular and immunotherapies have welcomed a new chapter in multi-modal cancer therapy. These agents harness our innate immune system and destroy malignant cells in a precise way as compared with "legacy" chemotherapeutic agents that largely rely on abolishing cell division. New therapies can augment the T-cell recognition of tumor antigens and effectively prevent tumor cells from their historically successful ability to evade immune recognition. These novel agents cause acute and chronic toxicities to a variety of organ systems (enteritis, pneumonitis, hypophysitis, and hepatitis), and this may masquerade as other chronic illnesses or paraneoplastic effects. As the perioperative footprint of cancer patients increases, it is essential that perioperative providers—anesthesiologists, surgeons, nurse anesthetists, and inpatient hospital medicine providers—be up to date on the physiologic mechanisms that underlie these new therapies as well as their acute and subacute toxicity profiles. Immunotherapy toxicity can significantly impact perioperative morbidity as well as influence perioperative management, such as prophylaxis for adrenal insufficiency, preoperative pulmonary assessment, and screening for thyroid dysfunction, among others.

**Keywords:** immunotherapy; cellular therapy; perioperative risk; hypophysitis; pneumonitis





## 1. Introduction—A New Paradigm of Perioperative Cancer Care

The last decade has brought about a revolution in cancer therapies that harness the immune system to target cancer-specific antigens. These agents were once reserved for use in palliative chemotherapy for patients with unresectable or refractory metastatic disease. They have now firmly established their intention-to-cure role and play a significant role in neo-adjuvant therapy [1–13]. Immunotherapies have greatly improved quality of life for patients with advanced cancer. More than ever, it is necessary for the perioperative provider to be up to date on the physiology underlying these therapies as well as their unique toxicity profiles.

Early-generation chemotherapeutics relied on nonspecific destruction of cell division, with toxicities mostly related to en masse destruction of rapidly dividing cells. Newer immunotherapies have a more benign acute safety profile but nonetheless can exert chronic toxicities that may be more insidious and require a higher index of suspicion for detection. The predominant cause of toxicities results from unwanted upregulated immune system activity or molecular mimicry, leading to destruction of "bystander" tissue.

To provide an example, ipilumumab and nivolumab are two immune-checkpoint inhibitors (ICIs) that have recently been shown to improve survival when included as part of neoadjuvant therapy for squamous cell esophageal cancer [14]. However, the two drugs in combination can lead to a non-negligible incidence of ICI pneumonitis and ICI hypophysitis. It is thus becoming more important for thoracic surgeons and anesthesiologists who care for these patients to be aware of the potential risks of adrenal insufficiency and reduced pulmonary capacity. A thorough therapeutic history and close collaboration with oncologists are necessary to provide safe care.

The purpose of this review is to give the perioperative provider physiological insight into new targeted immunotherapies, with a focus on their implications for safe perioperative care.

## 2. Immune-Checkpoint Inhibitors

Immune-checkpoint inhibitors have earned their keep as tremendously successful adjuncts to all manner of chemotherapeutic regimens. Our innate immune system relies on T-cell activation and proliferation to destroy foreign cells. T-cell activation requires both antigen–receptor coupling as well as co-stimulation by other immune effectors cells. This two-fold process underpins the versatility of our immune system but is also the process by which cancer cells can evade detection. T-cell activation can be inhibited by two pathways: the cytotoxic T-lymphocyte-associated antigen 4 (CTLA-4) pathway and the programmed cell death protein (PD-LA1) pathway. This interplay of T-cell activation and inhibition is referred to as an "immune checkpoint."

Cancer cells have evolved to thwart T-cell activation by taking advantage of the inhibition role of this immune checkpoint since CTLA-4 and PD-LA1 antigens are commonly found on the surfaces of tumor cells. Activation of these ligands by tumor cells allows them to evade detection.

Immune-checkpoint inhibitors (ICIs), as a therapeutic target, prevent tumor cell evasion by acting as monoclonal antibodies against the CTLA-4 and PD-LA1 ligands. In preventing their ability to inactivate T cells, ICIs ensure that tumor cells are vulnerable to destruction (see Figure 1).

This has proven to be a very effective therapeutic target, with successes in the treatment of melanoma, breast cancer, and lymphoma, to name a few.

Checkpoint inhibition technology has continued to advance as new agents are developed that target additional T-cell pathways and as existing ones are used on new tumor subtypes.

The first immune-checkpoint inhibitors approved for human use were Ipilimumab and Nivolumab (2011 and 2014, respectively). Ipilimumab acts as a monoclonal antibody against CTLA-4 and Nivolumab against PD-1. Ipilimumab was initially shown to dramatically improve survival in patients with metastatic melanoma [15]. The subsequent arrival of Nivolumab showed promise for the two used as combination therapy both for metastatic melanoma and several other cancers—renal cell carcinoma, unresectable non-small-cell lung cancer (NSCL), prostate cancer, and esophageal cancer [15,16].

Immune-checkpoint inhibitors have shown promise in treating tumors that have been historically resistant to other chemo or immunotherapeutic agents, but there are highly specific indications for the use of each. Up to 45% of patients with cancer are candidates for ICI therapy [17], so the perioperative footprint of these patients continues to rise. Their efficacy is determined by the rate of target PD-L1 or CTLA-4 receptor expression on the tumor cells, the mutational burden of tumors, and the relative inefficacy of classic chemotherapy agents for the tumor in question.

ICI therapy in the perioperative period is an area of ongoing study. Thus far, a few studies at major cancer centers have shown no association with perioperative morbidity of any type [18], and in fact, most show greatly improved results when added as neo-adjuvant therapy [9–11].

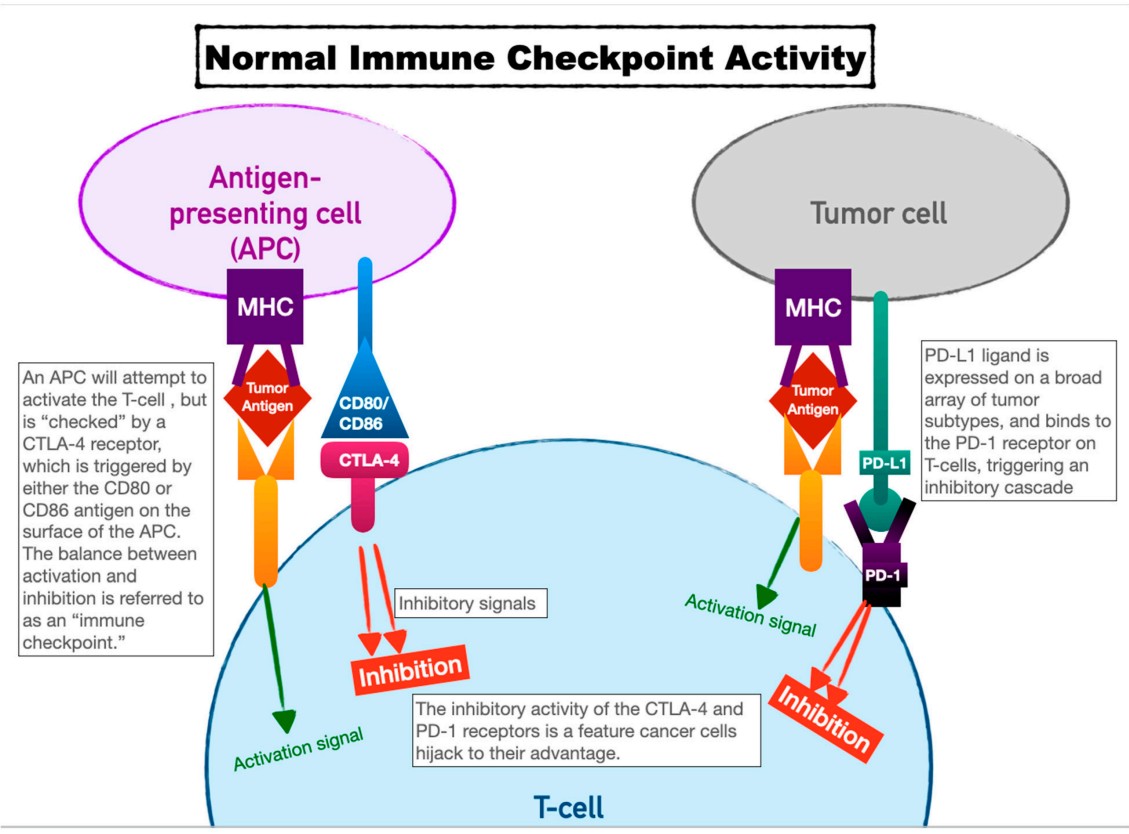

**Figure 1.** Above: normal immune checkpoint activity means that cancer (as tumor cells directly or as tumor antigen presented via an APC) can inhibit T-cell activation via the CTLA-4 and PD-1 ligands. Below: immune checkpoint inhibitors prevent this inhibition, leading to T-cell activation.

## 2.1. Specific Toxicities and Perioperative Implications

Toxicity from ICIs occurs via multiple mechanisms. The up-regulation of T-cell activity in general can lead to lymphocyte infiltration in normal parenchyma and cause nonspecific tissue damage. ICIs can lead to a loss of immunologic self-tolerance, wherein more T and B cells will be activated regardless of the presence of tumor cells. Molecular mimicry can occur, wherein ICI targets may appear on non-tumor cells, which are inadvertently triggered for destruction. Direct drug toxicity is less common owing to their relatively inert antibody structure [19].

### 2.1.1. Gastrointestinal

The increase in T-cell activation by ICIs is most clinically significant in sites that have a high number of T cells: notably the lining of the GI tract, the airways, and skin. As such, the most prevalent condition is immune-checkpoint-inhibitor-induced enterocolitis, which can occur in up to 20% of patients, and onset is typically within 2–3 months of therapy initiation. Mild cases of ICI enterocolitis (a few loose stools per day) are treated with low doses of oral corticosteroids. Endoscopy is typically performed for patients with an increase in stool of over 4–6 more events over their baseline to establish diagnosis and exclude other infectious or malabsorptive causes. These patients with moderate disease will be started on steroids and considered for biologic agents including infliximab. Repeat endoscopy is indicated after 2–3 months to gauge endoscopic response. For patients with mild symptoms, temporary cessation or low-dose corticosteroids might suffice. For moderate to severe symptoms, ICI therapy is usually stopped permanently [20]. Surgical intervention is extremely rare and reserved for cases of severe, refractory colitis with megacolon or perforation [21].

Perioperative implications of this condition track with disease severity. Patients on ICI should be screened for symptoms of gastritis that might affect anesthetic management, i.e., rapid sequence induction. Patients with malabsorptive diarrhea warrant a comprehensive metabolic panel and screening for electrolyte analysis.

ICI hepatitis is a less common entity but critical to not miss. It has an incidence of 5–10% [14,15], usually occurring within 2–3 months of therapy. It will typically manifest as an occult increase in liver function testing (LFTs). The pattern of liver dysfunction typically follows a hepatocellular pattern (AST and ALT elevation) and less commonly a cholestatic pattern (ALK elevation and hyperbilirubinemia). The risk increases with dual-ICI therapy [16,17]. CT imaging should be performed to rule out liver metastases, and abdominal ultrasound should be performed to rule out biliary obstruction in patients with hyperbilirubinemia. Workup should include hepatitis viral panels, history of alcohol use, and presence of concomitant hepatotoxins.

ICI hepatitis is graded on a scale: grades 1 and 2 are defined by AST and ALT levels of 2.5 to 5 times normal. These cases are typically treated with corticosteroids, which can be tapered over a 4–6-week period once LFTs return to normal. Very mild cases in patients with known liver metastases may warrant observation alone. Grade 3 and 4 ICI hepatitis are defined by a 5- to 20-fold elevation of LFTs and the presence of any clinical signs of liver dysfunction (asterixis, encephalopathy, and ascites). In addition to drug cessation and a course of IV corticosteroids (typically methylprednisolone 1 mg/kg/day), a liver biopsy may be warranted to confirm the diagnosis or evaluate for other causes of liver dysfunction. For patients with steroid-refractory liver dysfunction, mycophenolate mofetil and tacrolimus can be considered in coordination with a hepatologist experienced in treating the condition [22]. Once treated and resolved, ICI therapy is only resumed for patients who achieved improvement from grade 2 toxicity or less.

To mitigate perioperative risk, we recommend LFTs for patients requiring surgery on ICI therapy. LFT elevation should prompt additional workup including hepatitis viral panels, history of alcohol use, CT scan of the liver, and evaluation for concomitant hepatotoxic agents. Perioperative studies on patients with known ICI hepatitis are limited. It is recommended to postpone elective surgery on patients with hepatitis of any cause. Patients with ICI hepatitis should be treated as patients with liver dysfunction of any

other cause. The Child–Pugh classification of liver dysfunction has been validated as a classification system of patients with cirrhosis to distinguish surgical mortality, with an increased score assigned to patients with signs of decompensated hepatic failure, such as coagulopathy, ascites, and encephalopathy [23].

A less common effect of ICI therapy is pancreatic dysfunction. Approximately 4% of patients on ICIs will have an asymptomatically elevated lipase. The incidence of patients with symptomatic pancreatitis is <1% [24,25]. Symptoms of ICI pancreatitis are similar to those of all-cause pancreatitis, including epigastric pain, fever, and nausea. A CT scan should be obtained to rule out other causes of pancreatitis and biliary obstruction. Patients with asymptomatic lipase elevation are not recommended to receive any specific treatment other than monitoring. Symptomatic pancreatitis should be treated with usual care: fluid resuscitation, NPO status, and analgesia. It is notable that studies are extremely limited for ICI pancreatitis, including the lack of a standardized grading system. Most centers use steroids and ICI cessation for patients with severe symptoms requiring hospitalization [20–22]. As with usual pancreatitis, antibiotics are not routinely recommended in the absence of abscess or presence of another infectious source [26,27]. There is no established role for surgery in ICI pancreatitis, which in general is a last-resort intervention for complicated pancreatitis of all causes.

New-onset type 1 diabetes is a rare result of acute autoimmune pancreatic exocrine dysfunction, with an incidence of <1%. The most common presentation of this adverse effect is new-onset DKA. Destruction of islet cells renders patients permanently insulin-dependent in most cases, which is an outcome that is not treatable with steroids [28].

### 2.1.2. Pulmonary

Pulmonary toxicity from ICIs is referred to as ICI pneumonitis. The overall incidence is approximately 5% [29,30]. Cough and dyspnea are the typical initial symptoms, but the clinical syndrome falls on a wide spectrum of respiratory disease: acute, chronic, waxing/waning, and permanent disabling.

The acute form of ICI pulmonary toxicity follows a pattern of acute hypersensitivity pneumonitis and can progress to full-blown acute respiratory distress syndrome (ARDS). Chronic ICI pneumonitis occurs within 3 months to one year after initiation of therapy [31]. In both conditions, alternative causes of respiratory symptoms should be ruled out with a history of exposures, toxins, and smoking history, and CT scans with contrast should be obtained. Radiologic patterns of chronic pulmonary toxicity typically reveal ground-glass opacities and peripheral multifocal consolidations, although there is no pathognomonic finding for ICI pneumonitis. Bronchoscopy is only recommended for patients that do not respond to initial treatment or for whom alternative diagnoses are being ruled out, i.e., organizing infection, progression of malignancy, etc. Bronchoscopy specimens, if obtained, will show alveolar lymphocytic infiltration. Pathology specimens in ICI pneumonitis show a pattern of bronchiolitis obliterans and organizing pneumonia, with a lymphocytic predominance [32]. Smoking and prior radiation therapy increase the likelihood of ICI pneumonitis [33].

ICI pneumonitis is graded on a scale of grade 1 to 5 [34]. Patients with no symptoms and only mild radiographic findings are considered grade 1 and simply warrant observation with continuation of treatment. Patients with grade 2–5 (symptoms ranging from dyspnea/cough to fulminant respiratory failure) should be treated with glucocorticoids and cessation of treatment. Typical doses of prednisone range from 0.5–1 mg/kg daily, and the typical course is 2–4 weeks. Addition of anti-IL-6 agents has shown some promising results in a subset of patients who are refractory to steroids. The prognosis of ICI pneumonitis varies, as the pattern of disease is heterogenous. Patients with grade 1–3 disease do well with early recognition and steroid treatment. Patients with grade 4 or above or who have recurrent or treatment-refractory disease have a significantly higher burden of inpatient care, oxygen dependence, and overall mortality. It is worth noting that radiographic findings can persist long after symptom resolution [34,35].

Perioperative management of patients with ICI pneumonitis varies depending on the context in which it arises. For patients on ICI therapy as neo-adjuvant to surgery, such as in non-small cell lung cancer [36], the benefits of therapy have been shown to outweigh the risks. However, it remains essential to screen patients for pneumonitis symptomatology, which will inform operative risk. A history of dyspnea or cough in any patient on ICIs should warrant a chest X-ray, and their surgical risk should be stratified based on standard-of-care for patients with all-cause pulmonary disease. Recommendations for preoperative lung function testing are continually evolving, with recommendations including spirometry for all patients undergoing lung resection and for patients with a history of severe obstructive pulmonary disease. Patients with forced expiratory volume in one second (FEV1) and diffusion capacity of the lungs for carbon monoxide (DLCO) that are both >80% of predicted values are considered fit for surgery, while those with decreased scores may benefit from further workup. Stair climbing and ambulatory stress tests (such as walking 400 m on level ground with dyspnea and pulse oximetry monitoring) have been evaluated, and there is increasing evidence supporting the use of cardio-pulmonary exercise testing for further risk stratification [37]. While further studies on outcomes specific to ICI pneumonitis are lacking, it is prudent to follow these existing guidelines.

### 2.1.3. Cardiac

ICI myocarditis is rare, with an incidence of less than 0.1% [38]. While low, the mortality is significantly higher than other ICI toxicities, and in cancer centers with a high relative volume of immunotherapy patients, its prevalence is not to be ignored. Toxicity mechanisms are unclear but likely result from lymphocytic infiltration of myocardial tissue in the setting of upregulated T-cell activity. Pembrolizumab is the most common causative agent; the incidence is higher when combined with nivolumab. The average time to onset of cardiac symptoms is 1–2 months.

Cardiac toxicity can present as a conduction system disorder (ranging from prolonged PR interval to complete heart block), pericarditis, myocarditis, or cardiomyopathy with reduced systolic function [39,40]. A useful triage of myocarditis in the presence of cancer therapeutics was developed by Circulation in 2019, grouping patients into three categories: definite myocarditis, probable myocarditis, and possible myocarditis. A positive cardiac MRI plus one of either (1) an elevation in biomarkers, (2) an ECG pattern showing diffuse ST elevations, or (3) the clinical syndrome of myocarditis are required for the diagnosis [35–37].

Mild cases of ICI myocarditis are treated with oral prednisone. More severe cases may require intravenous methylprednisolone in high doses, which mimics the treatment strategy of viral myocarditis. A taper regimen will generally last 4–6 weeks. Treatment with intravenous immunoglobulins, mycophenolate, infliximab, and plasmapheresis has been considered for severe cases and remains under investigation [39].

In addition to steroids, it is recommended to follow the standard American College of Cardiology/American Heart Association guidelines for the treatment of heart failure: beta blockade (weighed against the risk of exacerbating a concurrent conduction system abnormality), ACE inhibition, lifestyle modifications, and avoidance of volume overload.

Perioperative considerations for ICI myocarditis include a thorough cardiac history and a recent EKG to evaluate for conduction system abnormalities, particularly with a history of pembrolizumab exposure. There have been several case reports of complete heart block noted on a baseline preoperative EKG among patients on checkpoint inhibitors planned for cancer resection surgery. Patients with dyspnea on exertion, chest pain, or unexplained hemodynamic compromise should be evaluated with an EKG, cardiac biomarkers, and an echocardiogram. Suspicions of myocarditis should prompt evaluation with a cardiologist and consideration for a cardiac MRI.

### 2.1.4. Endocrine

Pituitary dysfunction is a well-established category of ICI toxicity. The incidence of hypophysitis varies depending on the agent, the studied population, and combination

agent but can range from 1–15%. The highest incidence occurs in patients on dual-ICI therapy with ipilimumab and nivolumab. A recent trial examining the efficacy of the ICI agent Veliparib in triple-negative breast cancer showed a very low risk of this adverse event [41,42]. The most common biochemical disturbances are ACTH deficiency, hypogonadism, and hypothyroidism. For patients not detected biochemically, common presenting symptoms are headache and fatigue, but anorexia, nausea, dizziness, and visual changes have also been described. Pituitary dysfunction typically presents 2–3 months after therapy initiation, but onset can be as late as one year into therapy. In contrast to other adverse ICI effects, hypopituitarism is often permanent.

Patients on ICI therapy have regular thyroid function studies as part of maintenance therapy. Clinical signs of hypothyroidism include headache and fatigue, and laboratory findings include low–normal free T4 and inappropriately low or normal TSH.

A substantial portion of patients on ICI therapy have reduced ACTH and cortisol. Retrospective studies of ICI therapy cohorts have reported a wide range of incidences of low hormone values, ranging from one-quarter to one-half of patients. Most are associated with CTLA-4 inhibitors, but PD-1 inhibitors will also cause isolated ACTH deficiency [41–43].

Low serum cortisol resulting from central adrenocortical suppression can result in hypoaldosteronism with hyponatremia and hyperkalemia. It should be noted that hyponatremia is an extremely common adverse event in patients on any chemotherapeutics, including as a result of cancer-related SIADH [43].

Hypogonadism on ICI therapy has an incidence of 10–15% and presents with decreased libido, hot flashes, or infertility. Laboratory studies will reveal low testosterone and inappropriately normal luteinizing hormone (LH) [44].

MRI findings in patients with ICI hypophysitis demonstrate pituitary enlargement in slightly more than half of cases; however, MRI findings do not necessarily correlate with clinical symptom burden [45,46], and patients with significantly reduced ACTH or cortisol may have normal MRI findings. Pathology specimens for patients with this condition indicate lymphocytic infiltration and increased CTLA-4 antigen expression on pituitary endocrine cells (i.e., molecular mimicry) [47] as the culprit pathology. Ectopic CTLA-4 has been found to be expressed on adenohypophyseal cells.

Hyperthyroidism has a lower incidence (3–8% [47,48]) and can result both from a centrally mediated increase in TRH from hypophysitis or a transient, local thyroiditis from molecular mimicry. Patients with locally mediated ICI thyroiditis will most commonly present with laboratory abnormalities (low TSH and elevated T3, T4) or present with symptoms of hyperthyroidism. Thyroiditis, after it subsides, will most often progress to hypothyroidism.

The perioperative provider should have a high index of suspicion for adrenal insufficiency. There are no studies thus far on the incidence of intraoperative hemodynamic instability related to ICI therapy, but anecdotal evidence exists for patients on ICI therapy to require higher doses of intraoperative vasopressors. There should be a high index of suspicion for detecting underlying central adrenal insufficiency and to treat empirically should hemodynamic instability occur. Hydration status should be optimized, and electrolytes should be checked pre- and postoperatively for all patients on ICI therapy. Adrenally insufficient patients are more likely to decompensate in the setting of systemic infection and benefit from early, aggressive resuscitative strategy, as they have limited ability to compensate for physiological stress (Table 1).

**Table 1.** Common manifestations of immune-checkpoint inhibitor toxicity and relevant perioperative implications.

| Immune-Checkpoint Inhibitor Toxicity Category | Manifestations | Treatment | Perioperative Considerations |
|---|---|---|---|
| Central hypothyroidism | Fatigue, headache, low T4, low/normal TSH | Thyroid hormone replacement | Thyroid function tests within 3 months for all patients on ICI therapy |
| Hypogonadism | Hot flashes, decreased libido, decreased FSH and LH | Hormone replacement therapy | Screening for other signs of pituitary dysfunction |
| Pulmonary | Hypersensitivity pneumonitis, pulmonary fibrosis, organizing pneumonia, ARDS | Corticosteroids and supportive care | Screening for respiratory symptoms, baseline CXR, pulmonary risk stratification with baseline spirometry and cardio-pulmonary exercise testing |
| Enterocolitis | Diarrhea, gastritis, hepatitis (manifests as occult rise in AST/ALT over first few months of therapy) | Low-dose corticosteroids | Screening for reflux, gastritis symptoms; recent LFTs if therapy was started within the last few months |
| Pancreatic dysfunction | New-onset type 1 diabetes and ICI pancreatitis | Corticosteroids, resuscitative fluids, insulin therapy | Amylase and lipase levels for patients reporting abdominal pain, ruling out other causes of abdominal pain or pancreatitis; screening with preoperative blood glucose |
| Cardiac | Myocarditis, pericarditis, conduction system disorders | Corticosteroids and standard AHA/ACC heart failure guidelines | Screening for symptoms of heart failure, and baseline EKGs for patients on ICI therapy, cardiac biomarkers, and cardiac MRI for patients with symptoms of myocarditis |

### 3. Chimeric Antigen-Receptor T Cells (CAR-T)

CAR-T cells harness the adaptive immunity offered by our T-lymphocytes to target tumor-specific antigens. As the first foray into cellular therapeutics, they represent one of the most consequential discoveries in cancer therapy in the last two decades owing to their relative ease of administration and robust efficacy against highly fatal hematologic malignancies. As discussed in the previous section, the immune system relies on T-cell activation via (1) antigen presentation via an MHC complex and (2) co-stimulation via other immune effector cells. Cancer cells have historically evaded this process of T-cell activation by thwarting antigen recognition. Enter CAR-T cells: the genetically modified host T cells that force this process to occur without the ability of tumor-cell evasion.

First, T cells are extracted from the patient (or an allogeneic donor) and modified via a retroviral vector to express a new antigen receptor and co-stimulatory domains (see Figure 2). The term "chimeric" refers to the fact that this engineered receptor both activates and co-stimulates the T cell [49,50]. The patient is then leuko-reduced to increase the relative proportion of CAR-T cells, and CAR-T cells are infused into the patient. The CAR-T cells will recognize antigens and activate and recruit immune effector cells to destroy tumor cells.

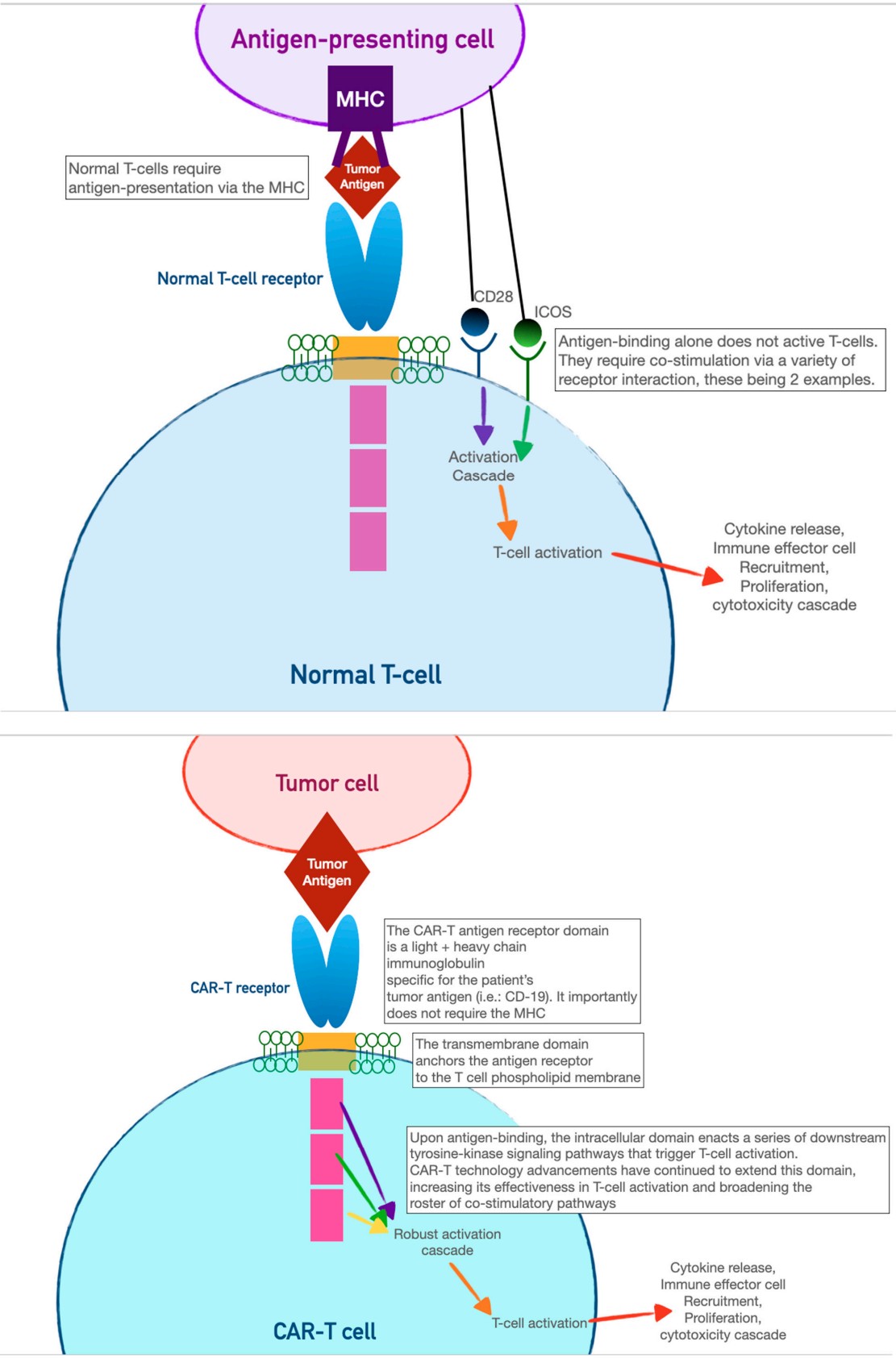

**Figure 2.** Above: normal T-cell activation requires antigen recognition and co-stimulation. Below: a re-engineered CAR-T cell enables simultaneous antigen recognition and co-stimulation, leading to T-cell-mediated destruction of tumor cells.

The first CAR-T cells used clinically targeted the CD19 receptor, making them effective for the treatment of ALL or diffuse large B-cell lymphoma [51]. Trials have consistently shown a 50–90% remission rate for relapsed or refractory disease, which continues to improve with refinement of technique [52]. CAR-T for solid tumors is a growing area of research, where re-modeled CAR-T cells can be introduced intra-tumorally or intra-pleurally [53].

Once infused, CAR-T cell activity can be controlled via embedded biochemical switches. An "on" switch or, conversely, a "suicide gene" can induce CAR-T cell activation or apoptosis. These external controls allow CAR-T technology to be more pliable in the face of toxicities or other adverse events.

## 4. Bispecific T-Cell Engager (BiTE) Therapy

BiTE therapy is one of the newest innovations in targeted immunomodulatory chemotherapy. BiTE molecules are an antibody with two domains (hence "bispecific"): one that recognizes tumor-specific antigens and another that recognizes the universal CD-3 receptor on T cells (see Figure 3). The binding sites are essentially two monoclonal antibodies bound together. In essence, the BiTE "forces" a recognition and activation of the T cell, so the tumor cell does not have a chance to present its own antigen and possibly release inhibitory signal receptors. Specific tumor antigen targets of BiTE molecules include CD19 (broadly expressed on B-cell malignancies), B-cell maturation antigen (expressed highly on malignant cells involved in multiple myeloma), and CD33 (expressed in acute myeloid leukemia, myelodysplastic syndrome, and chronic myeloid leukemia). Under development are BiTE targets that include solid tumor antigens such as prostate-specific member antigen and delta-like protein 3, which is highly prevalent in small-cell lung cancer.

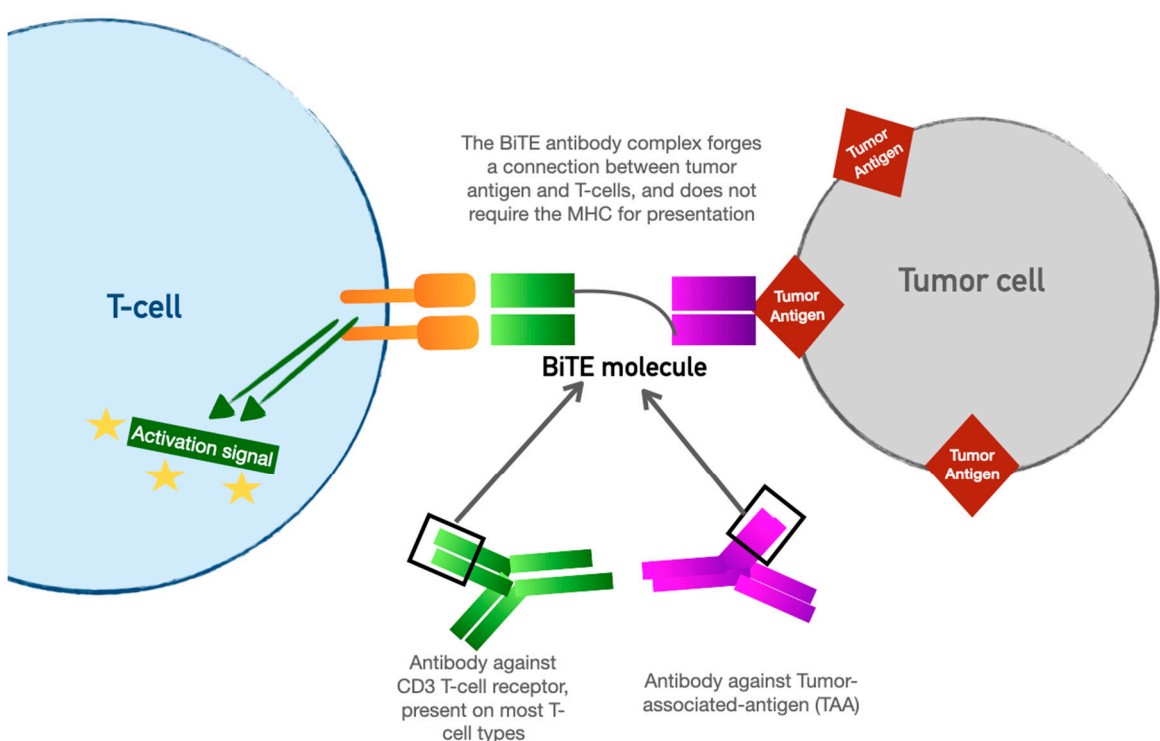

**Figure 3.** The Bispecific T-cell engager (BiTE) molecule has shown promise as a targeted immunotherapy but, unlike CAR-T, does not require harvesting of the patient's native lymphocytes.

The most widely used BiTE agent at this time is Blinatumomab, approved for use in acute lymphoblastic leukemia (ALL). Trial response rates in children and adults with ALL have shown a resounding 90% complete remission rate, with a 6-month event-free survival of 67% and a 78% overall survival rate [54]. It is now approved for use around the globe. Solitumab is a BiTE molecule with a binding domain for the epithelial cell adhesion molecule (EpCAM), which is being investigated for therapy against gastrointestinal, lung, and other solid tumors [55].

These molecules are administered as continuous IV infusion over several days owing to their 2–4-h half-life. Not unsurprisingly, owing to their similar mechanisms, they have an adverse event profile that parallels that of CAR-T cells. In the original study for Blinatumomab, the most common adverse events were neutropenia, infection, elevated LFTs, neurotoxicity, and CRS [56]. CRS, neurotoxicity, and acute anaphylaxis are the most significant for the acute care and perioperative provider. There is perhaps a lower propensity than CAR-T for high-grade neurotoxicity [57]. Treatment for these adverse events follows the same protocol: corticosteroids, IL-6 inhibition, and supportive care.

## 5. Other Cellular Therapies

Several novel cellular therapies are under investigation: natural killer (NK) CAR-T cells, engineered T-cell receptors, and tumor-infiltrated lymphocytes. The latter involves harvesting lymphocytes that have already successfully infiltrated tumor cells (isolated via biopsy or surgical resection) and proliferating and re-infusing them. The increasing armamentarium of cellular therapy agents may differ in specific mechanism, but their underlying goal of T-cell activation and propensity to cause CRS and neurotoxicity remains similar.

## 6. Toxicity of Cellular Therapy and Perioperative Considerations

CAR-T recipients are monitored as inpatients for acute toxicity. Immediate post-infusion reactions include anaphylaxis and anaphylactoid reactions, including transient vasodilation. The most clinically significant toxicities are sub-acute and include cytokine-release syndrome (CRS) and neurotoxicity. Chief considerations for these conditions are summarized in Table 2.

**Table 2.** Chief considerations for cytokine release syndrome (CRC) and neurotoxicity.

| Toxicity | Incidence | Symptoms | Mechanism | Evidence-Based Treatments | Investigational Treatments |
|---|---|---|---|---|---|
| **Cytokine Release Syndrome** | Up to 50% | Vasodilation, confusion, fever, headache | Systemic inflammation induced by mass T-lymphocyte activation | Corticosteroids, Tocilizumab, time | Siltuximab, Anakinra |
| **Neurotoxicity** | 3–5% | Encephalopathy, hypoactive delirium, lethargy, tremors, seizures (see ICANS scoring) | Central nervous system infiltration of lymphocytes | Steroids, Tocilizumab, anti-epileptic drugs, time | Siltuximab, Anakinra |

### 6.1. Cytokine-Release Syndrome (CRS)

Around half of patients who have received CAR-T cells will develop some form of CRS [57]. This systemic inflammatory response results from en masse activation of lymphocytes and CAR-T-induced cytokine release. Interleukin-6 (IL-6) is highly associated with CRS severity and can be useful when attempting to differentiate between CAR-T-related CRS and CRS related to sepsis. CRS can occur within hours, but the peak onset-time is 4–7 days postinfusion.

CRS is grading occurs on a scale of 1–5. Grade 1 is defined as mild systemic symptoms such as low-grade fever, headache, and myalgias. Grade 2 is defined by the requirement

of oxygen or pressor support. Grade 3 is multi-organ dysfunction requiring high-dose or multiple pressors. Grade 4 requires the presence of mechanical ventilation and advanced multi-organ dysfunction. Treatment for CRS is mostly supportive, and it is notable that with just time alone, patients will make a significant recovery.

Patients with CRS in the perioperative period require close collaboration with the oncology services. Nearly all patients with CRS will receive glucocorticoids to blunt the overactive immune response. Grade 3 and 4 CRS should be treated with tocilizumab. Originally approved as a disease-modifying agent for rheumatoid arthritis, tocilizumab is a monoclonal antibody that prevents IL-6 from binding to its receptor [58]. It is administered on a three-times-daily dosing schedule and is commonly combined with corticosteroids. Siltuximab is an anti-IL 6 monoclonal antibody that has shown safety and equivalence to Tocilizumab [59]. Siltuximab should be noted as a strong CYP-450 inducer. Anakinra is an IL-1 antagonist that has been explored as an off-label agent to mitigate CRS and CAR-T neurotoxicity [59,60]. It can also be used to treat hemophagocytic lymphohistiocytosis (HLH), a severe complication of CAR-T therapy in which lymphocyte over-activation leads to destruction of all cell lines, liver dysfunction, coagulopathy, fever, and rash [61].

It is important to note that none of the CRS-targeted therapies discussed here will significantly impact the efficacy of CAR-T cell therapy.

### 6.2. Neurotoxicity

Overall, 3–5% of patients who receive CAR-T cells will develop neurotoxicity [62]. Commonly referred to as immune-effector-cell-associated neurotoxicity syndrome (ICANS), the etiology of this condition is multifactorial and not yet fully understood. A likely early contributor is the breakdown of cerebral endothelial integrity by increased circulating cytokines, like the pathophysiology underlying septic encephalopathy [63,64]. The typical timeframe for onset of ICANS is 5 days postinfusion but can be delayed by up to 2 weeks. Neurologic symptoms that are specific to early ICANS include word-finding difficulty, tremors, hypoactive delirium, lethargy, and impaired handwriting – there exist several published ICANS scoring systems for clinical use. Moderate ICANS can progress to hyperactive delirium, myoclonus, obtundation, and ataxia. Severe ICANS can result in neurologic catastrophe including seizures, coma, and death. The median duration of this syndrome is 4 days. Mortality is related to the degree of their resultant critical illness—ARDS, malnutrition, degree of shock, etc. [65,66]

Neuroimaging and video-electroencephalography (vEEG) obtained from ICANS patients will demonstrate nonspecific patterns of cerebral edema and generalized slowing, respectively. Lumbar puncture will reveal elevated lymphocytes in CSF as well as the presence of CAR-T cells.

Time and supportive care are the mainstay of treatment. ICANS is less responsive to the typical IL-6-targeted therapies as compared to CRS. Corticosteroids are beneficial but come with significant iatrogenic harm—immunosuppression, gastrointestinal bleeding, and increased delirium. There is some evidence showing seizure prophylaxis with levetiracetam to be useful for anyone with grade 2 or higher ICANS.

CRS and ICANS mortality are more closely linked to the degree of organ dysfunction and iatrogenic exposure (prolonged ICU stay, mechanical ventilation, pressor use, prolonged steroid use and immunosuppression, etc.) than the degree of the toxicity per se. With supportive care, a relatively high proportion of patients experience complete recovery [66,67]. The overall mortality of CAR-T cells remains low at 5.4% [67].

### 6.3. Perioperative Considerations

The increasing efficacy and safety profile of cellular therapies has placed them at the forefront of cancer therapeutics at a rapidly expanding array of academic and non-academic centers. At this time, around 150 centers across the country offer CAR-T therapy, and the technology has spread across the globe to Canada, China, Australia, Singapore, Israel, and Europe. Evidence for specific perioperative treatment is lacking, as are studies on outcomes

for emergent surgery performed in the immediate post-CAR-T period. The central tenets of safe perioperative care of such patients are (1) communication with the oncology team, (2) mitigation of infectious risk, and (3) preparation for hemodynamic compromise. It is crucial to involve the critical care team for patients with high-grade CRS or who are at risk for developing high-grade CRS for advanced cardiopulmonary monitoring.

Centers that offer CAR-T therapy are always equipped with services that are familiar with post-infusion monitoring and downstream complications such as CRS and neurotoxicity. For post-infusion patients with no signs of CRS, neurotoxicity, or infectious complications, usual care should be pursued for urgent or emergent surgery. Elective surgery should be deferred until hematologic dyscrasias have normalized, typically by 3–4 weeks. A discussion with the oncology team about intraoperative steroid dosing should be considered. For patients within a week of CAR-T infusion who need urgent surgery, there should be a low threshold for pre-emptive postoperative ICU admission so that acute changes in hemodynamics and neurologic status will not be missed. Coagulation profiles should be obtained prior to neuraxial anesthesia, and if no thrombocytopenia or elevated INR is present, there is no contraindication to regional anesthesia.

The physiologic milieu of CRS will render patients sensitive to the vasodilatory effects of anesthetics. They should be treated as patients with severe sepsis, with slow dosing of induction agents along with supplemental vasopressor support. Widespread endothelial dysfunction will lead to increased capillary leak, which portends to intravascular depletion and increased third spacing. Fevers and prolonged NPO status (in the setting of CRS or neurotoxicity) will exacerbate this. Insensible losses will be higher, so fluid-sparing strategies should be used with caution. As with any patient experiencing compromised end-organ perfusion, attention should be paid to not further worsen tissue oxygenation. Hypoxemia and hypercarbia should be avoided. Sedation without a protected airway should be administered cautiously in the presence of metabolic acidosis, and there should be a low threshold to provide an endotracheal airway. The presence of severe acute kidney injury (AKI) is uncommon in CRS, and most patients recover to their baseline renal function [67]. In such cases, dosing of paralytic and analgesics should be undertaken with attention to GFR.

Neurotoxic patients should be treated as at-risk for having elevations in intracranial pressure. Caution should be taken to avoid exacerbations of ICP (hypercarbia, hypoxemia, and Trendelenburg position), normo-glycemia, normothermia, and normal sodium homeostasis. Anticonvulsants should be continued. The ICANS scoring system should be used pre- and immediately postoperatively, so post-procedure neurologic changes can be evaluated [57].

## 7. Conclusions

Targeted cancer therapy is undergoing a seismic shift from one that exerts an imprecise attack on cell division to one that targets specific tumor antigens, engaging our innate immune system to assist in the process. With these more precise and efficient therapies comes an increasingly complex toxicity profile. Toxicities can occur chronically and masquerade beneath other neoplastic and paraneoplastic conditions. These agents increasingly play a role in multimodal cancer care, which is significantly prolonging life expectancy and quality of life for patients with cancer. As the depth and breadth of cancer procedures increases and as the footprint of these therapies increases, the perioperative provider plays a more central role in managing cancer therapies and drug toxicities. The above review serves to update the perioperative clinician on guidelines for screening, diagnosing, and treating toxicities associated with novel cellular therapies and immune-checkpoint therapies.

**Author Contributions:** All authors contributed to the conceptualization, original draft preparation, review, and editing. All authors have read and agreed to the published version of the manuscript.

**Funding:** This research received no external funding.

**Conflicts of Interest:** The authors declare no conflict of interest.

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
