# Peer review of "Novel Cellular and Immunotherapy: Toxicities and Perioperative Implications"

_curroncol, doi:10.3390/curroncol30080554_

Round 1

Reviewer 1 Report (Previous Reviewer 3)

The authors have made some improvements to the MS, but it is still not sufficient.

The MS title is incomprehensible and does not not match its content. The definition of the term "perioperative" is not clear. The treatments described can be done with or without surgery. This MS is about immunotherapy. It might be, for example, titled “Anticancer Immunotherapies and Immune-Mediated Complications”.

The abstract needs to be greatly reduced.

The MS needs to have a more clear structure and presentation.

Author Response

Thank you for your comments and for the time you took to review my paper. I have changed the title to make it more relevant and i have improved the length of the abstract, making it more relevant. 

Reviewer 2 Report (Previous Reviewer 2)

The current version of the article has shown significant progress in creating a focused article for clinical use, namely something that perioperative teams can read and be better informed of the unique needs of immune therapy patients undergoing surgery. I quite like the structure and much of the information contained there in.

I do still find several flaws. an annotated manuscript is attached for more detailed review. In brief summary, the abstract is unreadable and does not reflect the actual good usuable content in the article. It should be completely re-written to give a concise overview of the information that one would learn in reading the full article. ITs also much too long. Some of this info could be moved to the intro which is lacking a paragraph to ground the clinical context of the article. Within the substance of the article, there are various instances where the authors make statements with authority but in actuality they are not univerisally accepted practice. I think, given the dearth of published data in the field, its quite useful to give their institutions approach on preop screening, but the reader should know that that is what various statements are - institutional recommendations and not statements of fact. Also in various areas, the authors need to tighten references and also not select the most dramatic reference while not referencing more conservative numbers. Finally section 6 has an excellent summary of what these novel therapies are, but need to tighten the section on perioperative considerations to better reflect that outside of a phase 1 trial in very select institutions worldwide, these patients universially are only haveing surgery for an acute complication or illness and not as a planned event.

English language is good, though abstract is not really an abstract, but an introduction. needs re-write to be an accurate concise summary of the article.

Author Response

Thank you for your comments, i am grateful for the time you took to review! I totally agree with your feedback, and have made the suggested track changes. I agree with you that there are not established guidelines for perioperative CAR-T patients, given that surgery in this population is a rare and usually emergent event. However the prompt for this paper was specifically “perioperative” considerations, so i am working with what we have. Regarding shortening the recommendations on perioperative considerations for CAR-T cell patients: multiple reviewers expressed there was not enough perioperative insight, and that perioperative recommendations were the entire purpose of the requested topic, so i am worried about shortening what little i have. I think the physiologic insights might be interesting to some readers, as “food for thought” from a provider experienced in this rea. I re-did the abstract. i tightened the references. 

Reviewer 3 Report (Previous Reviewer 1)

The authors have successfully addressed the comments and the questions raised by the reviewers and the manuscript is significantly improved. I found it acceptable and publishable in the journal in its current format.

Author Response

Thank you for your comments, i am grateful for the time you took to review! 

Round 2

Reviewer 1 Report (Previous Reviewer 3)

My previous comments did not receive any responses. The title of the MS is still unfortunate. The authors shortened the abstract, but made it uninformative.

The writing in the MS is not well-done. For example, Figure 3 has no name and no caption. The tables are also unnamed.

The MS is unevenly and not always logically divided into sections and subsections. For example, section 6 “Other Cellular Therapies” includes only a few uninformative sentences. 

Author Response

Thank you for your comments and dedicated time to review my submission! I revised the abstract. I added a caption to figure 3, as you suggested. Regarding the “unfortunate” title, I revised it. I had elaborated more on “other” cellular therapies, but in a previous review, I was requested to shorten it, so that is why it is concise. I think that while it may be short in length, it gives the reader a sense that these therapies are rapidly expanding. For example, at our institution there has been confusion about NK-CARs and TILS, but my review suggests that their mechanisms are similar to CAR-T and thus their toxicities are similar. I think readers might like to hear what “cutting” edge therapies are on the docket. However, i am more than happy to eliminate this section if you find it to be superfluous!

Reviewer 2 Report (Previous Reviewer 2)

thank you for considering my suggestions. This is ready for publication. As someone who has done significant work in this area creating protocols for our institution, this is a useful overview for others who want to know more about ICI related periop care.

Author Response

Thank you for your comments!

Reviewer 3 Report (Previous Reviewer 1)

The authors have successfully addressed my previous comments and questions and the manuscript is now acceptable in the journal. 

Round 3

Reviewer 1 Report (Previous Reviewer 3)

The authors have markedly improved the MS. However, serious observations remain.

  1. The MS title is still unacceptable both semantically and linguistically. An acceptable title could be, for example, “T cell targeted, antitumor immunotherapy: toxicities and implications”.

  2. It remains unclear what the words "perioperative", "perioperative implications", and "perioperative suggestions" mean. Please either clarify the word "perioperative" or remove it from the text.

  3. The subsection 5 "Other Cellular Therapies" is not informative. It should be removed from the text.

This manuscript is a resubmission of an earlier submission. The following is a list of the peer review reports and author responses from that submission.

Round 1

Reviewer 1 Report

This is an interesting, informative, east to follow and understand review article about the new cancer therapies in human malignancies, and to this reviewer, it would be a very good source of information for a broad range of readers including cancer scientists, oncologists, surgeons, and molecular biologists. 

Reviewer 2 Report

perioperative considerations for newer cancer therapies is a very valid topic for a review article. The authors do seem knowledgeable about the agents. I found this a very hard article to read and will suggest a number of revisions. 

I think we need to start by considering the audience for such an article. By its title, the article is intended to give education for those managing periooperative care. The main audience therefore should be surgeons and anesthetists, while aknowledging other health care professionals may also benefit. From an oncologist perspective, there is benefit to eudcating the medical oncology community on what side effects and issues need to handed over well to perioperative teams that may not be familiar with these agents and their side effects. In this lens, I would dramatically shorten the physiologic description of each therapy and expand the sections on side effects. There needs to be clearer sections on management of complications that might be seen and managed in the perioperative team. There needs to be clearer instruction on when not to operate, or how to temporize things when surgery is emergent and can not wait. While there isn't a tonne of literature in the field of perioperative management, there is some and the article doesn't really touch much on it. Overall, this is how I think the review needs to be restructured.

For additional comments, the abstract is unreadable and does not give a clear and concise synopsis of what the article wishes to cover. Similarly the intro could be tightened to give a more concise sense of what the article wants to teach. I would order the immune check point inhibitors first and this will be the largest group of patients who will undergo surgery at some time point. The final two treatment groups could be summarized in a small paragraph together as there seems to be no information on discrete novel side effects that need special attention. It should be recognized that a patient with a cytokine release syndrome is only getting surgery if they have an immediately life threatening issue (things like cerebral bleeds or perforated bowels), and as such they will still be actively managed by their hematologic team in a tertiary care setting. IS there are data on outcomes of emergency surgery in this setting? If so, it should be discussed.

Finally, there are mutliple times where words have inappropriate hyphens in them. These should be reviewed and edited out.

Reviewer 3 Report

This review describes the main immunotherapeutic treatments available, such as Chimeric antigen-receptor T Cells (CAR-T), Immune checkpoint inhibitors, Bispecific T-cell engager (BiTE) therapy, and Cancer vaccines, that are well-known for their use in treating cancer. The review does not provide any new information as it does not present any new perspectives or interpretations of the data. There is a discrepancy between the title of the MS and what it contains. Unclear and undefined is the perioperative period and the treatments given during this time. The abstract is essentially a reiteration of the introduction. In general, the article is written extremely carelessly. The entirety of the “Cancer vaccine” section is comprised of just one reference (39).  The majority of the figures and tables are not labeled with numbers nor do they have captions.